# The Philadelphia Chromosome, from Negative to Positive: A Case Report of Relapsed Acute Lymphoblastic Leukemia following Allogeneic Stem Cell Transplantation

**DOI:** 10.3390/medicina59040671

**Published:** 2023-03-28

**Authors:** Elena-Cristina Marinescu, Horia Bumbea, Iuliana Iordan, Ion Dumitru, Dan Soare, Cristina Ciufu, Mihaela Gaman

**Affiliations:** 1Department of Methodology and Scientific Research, “Carol Davila” University of Medicine and Pharmacy, Eroilor Sanitari Boulevard, No. 8, 050474 Bucharest, Romania; 2Department of Hematology, Emergency University Hospital of Bucharest, Splaiul Independentei, No. 169, 050098 Bucharest, Romania; 3Department of Medical Semiology and Nephrology, “Carol Davila” University of Medicine and Pharmacy, 050474 Bucharest, Romania; 4Department of Hematology, “Carol Davila” University of Medicine and Pharmacy, Eroilor Sanitari Boulevard, No. 8, 050474 Bucharest, Romania

**Keywords:** acute lymphoblastic leukemia, Philadelphia chromosome, relapse, allogeneic stem cell transplantation

## Abstract

Relapsed acute lymphoblastic leukemia (ALL) represents a continuous challenge for the clinician. Despite recent advances in treatment, the risk of relapse remains significant. The clinical, biological, cytogenetic, and molecular characteristics may be different at the time of relapse. Current comprehensive genome sequencing studies suggest that most relapsed patients, especially those with late relapses, acquire new genetic abnormalities, usually within a minor clone that emerges after ALL diagnosis. We report the case of a 23-year-old young woman diagnosed with Philadelphia chromosome-negative B cell acute lymphoblastic leukemia. The patient underwent allogeneic stem cell transplantation (allo-HSCT) after complete remission. Despite having favorable prognostic factors at diagnosis, the disease relapsed early after allo-HSCT. The cytogenetic and molecular exams at relapse were positive for the Philadelphia chromosome, respectively for the Bcr-Abl transcript. What exactly led to the recurrence of this disease in a more aggressive cytogenetic and molecular form, although there were no predictive elements at diagnosis?

## 1. Introduction

Acute lymphoblastic leukemia (ALL) is the most common childhood cancer, but is less common in adolescents and young adults (AYA) and is rare in older adults. AYA is an age between 15 and 49 years, depending on the study. Patients with B precursor ALL-AYA are known to have worse outcomes and more treatment-related toxicities compared with children. The 5-year survival rate of ALL is higher than 90% in children [1], but falls significantly to 60–85% in adolescents and young adults ([2], pp. 3660–3668) and to less than 30% in older adults [3].

In terms of clinical and biological characteristics and response to treatment, AYA-ALL differs from the pediatric ALL population. Lower event-free and survival rates in AYA patients are due in part to unfavorable tumor biology. Although survival of adolescents and young adults with acute lymphoblastic leukemia has improved with the use of pediatric-inspired protocols [4], the results of those who relapse remain poor.

The Philadelphia chromosome (Ph) is the most common cytogenetic abnormality in adult patients with acute lymphoblastic leukemia, occurring in about 20% to 30% of all cases [5]. The incidence increases with age and represents the most common form of ALL in the elderly population and is therefore a relatively rare event in AYA (less than 20%) [6]. Patients with Ph-positive ALL are at increased risk for central nervous system (CNS) involvement and an aggressive clinical course. Historically, they have had worse outcome compared to their Ph-negative counterparts [7]. However, the outcomes of patients with Ph-ALL have changed dramatically with the addition of tyrosine kinase inhibitors (TKI) to cytotoxic chemotherapy, increasing the rate of complete response to as high as 90% [8].

Risk stratification by cytogenetics and molecular genetics reflects the prognostic heterogeneity of ALL that determines which patients are at a high risk of relapse and should be considered for more intensive treatment strategies, including allogeneic stem cell transplantation.

Also, minimal residual disease (MRD) assessment plays a central role in risk stratification and treatment guidance in patients with ALL. MRD assessment is of extreme prognostic importance, as MRD is one of the most important risk factors for relapse, as confirmed by multivariate prognostic analyses in many studies [9]. Measurable residual disease is the amount of cancer cells that remain after therapy. MRD is not a classic biomarker, but a measure of actual disease burden. Patients with morphologic complete remission (CR) may also have MRD. Laboratory techniques for MRD testing in ALL provide a higher degree of sensitivity and specificity than morphology. Current MRD testing includes flow cytometry (FC), polymerase chain reaction (PCR), and more recently, next-generation sequencing (NGS). MRD assessment by FC has a sensitivity of 10^−4^ (standard) to 10^−5^, by PCR a sensitivity of 10^−4^ to 10^−5^, while the NGS method has the highest sensitivity, at 10^−6^.

Various studies show that the lower the level of MRD before HSCT and the better the results, the lower the risk of relapse [10]. However, it is still unclear what is the best treatment approach in case of MRD positivity. Early systemic relapses, occurring within the first 12 months after allogeneic hematopoietic stem cell transplantation, have a poor prognosis with a median survival of 7.4 months and a long-term survival rate of only 20–30% [11].

Although therapy for ALL has made great strides in recent years, relapses are still present in a high percentage of cases and present a challenge to the clinician. Identification of predictive risk factors for relapse is necessary at diagnosis and it is also necessary to evaluate them during the development and treatment of the disease.

The case of ALL, presented below, represents a rarer situation of early relapse after bone marrow transplantation, involving the occurrence of a cytogenetic change with an unfavorable prognosis that was not present at diagnosis. It was a challenge to identify the possible factors that could have caused this rather atypical relapse(Figure 1).

## 2. Presentation

We discuss the case of a 21-year-old woman who came to the emergency department with altered general condition, fatigue, loss of appetite, cough, fever, and purpura. Her medical history was irrelevant. Laboratory findings on admission revealed life-threatening anemia (a hemoglobin of 2.3 g/dL), severe thrombocytopenia (a platelet count of 24 × 10^9^/L), and a normal leukocyte count of 6.8 × 10^9^/L. In the peripheral blood smear (PBS), we found 6% medium-sized blasts with a morphological aspect suggestive of lymphoblasts (ALL-1, according to the FAB classification). Bone marrow examination revealed involvement of 38% blasts similar to those described in PBS.

Immunophenotyping by flow cytometry from bone marrow aspirate identified 33% cells with the following aspect: SSC low, CD34+, cMPO-, cCD79a+, cCD3-, CD3-, CD7-, CD117-, CD13-, CD33-, CD10+ (CALLA), CD19+, CD20-, sIgM kappa, lambda-, CD66c- (KOR-SA), NG.2-, CD123-, cTdT-, CD58-. The described are based on the diagnosis of acute B-cell lymphoblastic leukemia, positive for CALLA, not otherwise specified, according to the 2016 World Health Organization (WHO) criteria.

Cytogenetic examination carried out on 10 metaphases revealed that there were no numerical or structural alterations in chromosomes at this level of resolution (it should be noted that the cellularity of the bone marrow aspirate was low). In addition, a molecular study was performed that did not detect the Bcr-Abl transcript or the MLL fusion gene.

At the time of diagnosis, the patient had no neurologic signs or symptoms. Cerebrospinal fluid (CSF) studies and imaging also did not describe central nervous system (CNS) involvement.

Taking into account the available data—young age, normal white blood cells, precursor of the B lineage, positive for CALLA (CD10+), no evidence of CNS involvement, and no Philadelphia chromosome—we classified the patient in a standard risk group.

Based on the preponderance of the data presented in the review, the best therapeutic approach for this patient would be intensive pediatric treatment. We decided to treat according to the Berlin-Frankfurt-Münster (BFM) protocol because we knew the following data from the literature. The BFM 2000 protocol consisted of the following stages: induction (prednisone, vincristine, daunorubicin, PEG -asparaginase, intrathecal methotrexate), early intensification (cyclophosphamide, cytarabine, 6-mercaptopurine, intrathecal methotrexate), consolidation (combination of dexamethasone, vincristine, vindensine HD-cytarabine, HD-methotrexate, cyclophosphamide, ifosfamide, PEG–asparaginase, etoposide, intrathecal therapy), and re-induction therapy (dexamethasone, vincristine, doxorubicin, PEG–asparaginase, cyclophosphamide, cytarabine, 6-thioguanine), followed by maintenance (6-mercaptopurine, methotrexate).

The patient achieved complete morphological marrow remission immediately after the induction phase. After the reinduction phase, i.e., 9 months after the start of treatment, the patient was still in complete morphologic remission, without CNS damage, but with the presence of minimal residual disease of less than 0.1% as determined by flow cytometry. It should be noted that no major infectious or hemorrhagic complications or other severe toxicities occurred during chemotherapy according to the BFM protocol, except for grade 2–3 hepatic cytolysis, which was controlled by hepatoprotective treatment.

At this time, the patient underwent allo-HSCT from a matched related donor, knowing that MRD is the most important prognostic information and decision support for allocation to allogeneic hematopoietic stem cell transplantation. The procedure was successful, with no major complications and no graft-versus-host disease (GVHD). Chemotherapy-based myeloablative transplant conditioning regimens were applied without total body irradiation (TBI) for technical reasons. Six months after allo-HSCT, the patient was in complete remission, with negative MRD and 100% chimerism.

After another three months, the patient came to the scheduled follow-up in good general condition. However, PBS revealed a recurrence of blast cells, which raised suspicion of relapse of the disease. We performed a reexamination to determine the current state of the disease. Morphologically, the blasts had a similar appearance to that at diagnosis (ALL-1). The bone marrow aspirate showed the presence of 85% lymphoblasts.

The immunophenotypic aspect of the bone marrow blast cells showed some changes from the time of diagnosis, namely the positivity of the CD66c marker (KOR-SA), indicating the likelihood of the presence of the Philadelphia chromosome (Figure 2). This suspicion was confirmed by the result of the cytogenetic and molecular examinations. The Philadelphia chromosome was detected in 34% of the analyzed 10 metaphases. In 45.1% of malignant cells, molecular analysis identified the Bcr-Abl e1a2 transcript (m-Bcr) corresponding to protein p190. An examination of the central nervous system was also performed, and no relapse was found at this level.

At this point, in the face of a relapse with the presence of the Philadelphia chromosome, we decided to initiate treatment with dasatinib, knowing that this TKI has the advantage of preventing CNS involvement. The selected dose of dasatinib was 70 mg/day. Given the corticosteroid sensitivity the disease exhibited at the time of diagnosis, dexamethasone therapy was also administered.

After only 3 days of TKI treatment, we observed the disappearance of blast precursors from peripheral blood. The occurrence of grade 2–3 liver toxicity was noted, which required the interruption of TKI treatment. Complete remission of liver toxicity occurred after 7 days of dasatinib treatment. Subsequently, TKI was administered again, but at a dose reduced to 50 mg/day. It should be mentioned that there were no other toxicities, either hematologic or nonhematologic of higher magnitude, that would require interruption of treatment.

Very quickly, after another week of treatment with dasatinib, bone marrow evaluation already showed morphologic remission and Bcr-Abl transcript was no longer detected. It should be noted that the sample studied was from peripheral blood and the number of copies was reduced.

At this time, the patient is on treatment with dasatinib 3 months after relapse after bone marrow transplantation, with maintenance of morphologic and molecular remission, and is awaiting CAR-T therapy as a bridge to a second bone marrow transplantation.

## 3. Discussion

AYA with ALL represent a special subset of patients compared with the pediatric or adult population.

Although the survival rate for most children with ALL is now 90%, older adolescents and young adults historically have a much worse prognosis, with an event-free survival rate (EFS) of only 30% to 45% [12].

Over the past decade, treatment outcomes for AYA patients have improved with a disease-free survival rate of 60% to 70% when treated with pediatric approaches [13].

The superiority of pediatric approaches for the AYA population has now been demonstrated in several retrospective analyses, with higher complete remission rates, higher event-free survival, lower risk of relapse, and with comparable rates of nonrelapse mortality.

However, the number of cases of AYA-ALL that relapse remains high and is a challenge.

This case of acute lymphoblastic leukemia with a precursor of the B lineage, Ph-negative, with early relapse less than one year after HSCT (although the initial features of the disease placed the patient in a low-risk group) raises several questions. There are certain biological characteristics that influence treatment response and survival.

In our case, the biological, morphological, immunophenotypic, cytogenetic, and molecular characteristics at the time of diagnosis were not predictive of relapse. Certain characteristics of blast cells might allow predictions of the presence of certain genetic or molecular alterations and of the prognosis. It is well known that the expression of the markers CD66c [14], CD33 [15] or CD13 [16] can be correlated with the presence of the Philadelphia chromosome. NG2 (neuronal glial antigen 2) also has predictive value for MLL rearrangements [17]. In our case, the absence of the Philadelphia chromosome, the BCR-ABL1 fusion gene, and the absence of the MLL gene was later confirmed by cytogenetic and molecular studies. The presence of CALLA provides a good prognosis for the disease.

On the other hand, the pattern of ALL cytogenetics at AYA differ from that of the pediatric population. For example, a newly recognized unfavorable risk entity, BCR-ABL1-like (Ph-like) ALL, describes a category of B cells in ALL characterized by gene expression profiles similar to those of BCR-ABL1 ALL but lacking BCR-ABL1 translocation [18]. The incidence of Ph-like ALL appears to peak in adolescents, with a prevalence of nearly 30% (compared with 10% in children and 20–25% in older adults), and Ph-like ALL is associated with a poor prognosis across the age spectrum [19].

Overall, the lower incidence of cytogenetic abnormalities associated with favorable outcome, combined with the higher proportion of patients with low-risk genetic features, such as Ph+ and Ph-like ALL, in AYA patients contributes to their poorer survival compared with children. In our case, this evaluation of the presence or absence of this genetic abnormality could not be performed.

The choice of treatment was made with knowledge of the data from the literature, which states that the patients with AYA-ALL treated with the BFM protocol had high overall survival (OS) and complete remission (CRD) rates and tolerated this pediatric regimen with acceptable toxicity. The OS and CRD rates were comparable to those achieved with hyper-CVAD, an adult leukemia therapy ([2], pp. 3589–3754)

The persistence of minimal residual disease at the end of chemotherapy led to the indication of bone marrow allotransplantation. When feasible, allogeneic HSCT is preferable to standard intensive pediatric chemotherapy in MRD-positive patients to reduce the risk of relapse and increase survival from ≤25% without HSCT to approximately 45–55% (GMALL, NILG, GRAALL studies, reviewed by Bassan et al.) [9].

However, 9 months after bone marrow transplantation, the patient relapsed with a new chromosomal change (Philadelphia chromosome) compared to the time of diagnosis, although 9 months after transplantation, minimal residual disease was negative and chimerism was 100%.

Measurement of donor chimerism using peripheral blood and/or bone marrow samples is an alternative method for detecting residual or impending disease recurrence. Assessment of donor chimerism may be particularly valuable in patients for early detection of impending relapse when a reliable MRD marker has been lost and/or cannot be determined after transplantation [20]. Data suggest that the relevance of low MRD positivity after allo-HCT depends on the time elapsed since transplantation. In a study of pediatric patients, the longer the time has elapsed from the time of transplantation, the higher the risk of treatment failure in the presence of low-level MRD [10].

Several hypotheses can be considered that underlie the relapse of the disease: Was there a dormant clone that could not be identified at diagnosis that escaped immune system control? Is this mutation acquired during the development of the disease or is it secondary to treatment? Could possible contamination by the donor also be considered? Did the use of a myeloablative conditioning regimen in TBI potentially favor relapse?

In view of all this, the question arises of whether it would not be more effective to include TKIs in treatment in prophylactic doses and after certain clinical trials, starting from the diagnosis of ALL.

Relapse after allo-HSCT in patients with ALL has a poor prognosis. However, in modern times, outcomes and management of this group of patients have improved. The advent of immunotherapy, deep insights into the molecular mechanisms leading to resistance, and the discovery of new targeted therapies have led to numerous clinical trials and provide new hope for this patient group.

## 4. Conclusions

The Philadelphia chromosome is the most common cytogenetic alteration associated with acute lymphoblastic leukemia. It leads to an unfavorable prognosis for the disease. However, the introduction and development of TKIs have improved response rate and survival in this disease. The Philadelphia chromosome may be present at diagnosis or may occur later in the course of the disease and contribute to relapse of the disease.

It is extremely important that, in the case of acute leukemia relapse but also throughout the evolution of the disease, a complete and thorough reevaluation, both morphological and immunophenotypic, as well as cytogenetic and molecular, is carried out, because there may be acquisitions of new changes compared to the time of diagnosis, which require other targeted lines of treatment.

## Figures and Tables

**Figure 1 medicina-59-00671-f001:**
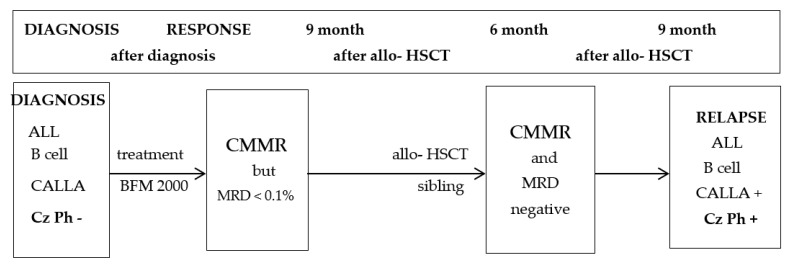
Evolution of disease. CMMR = complete medullary morphological remission; MRD = minimum residual disease; Allo-HSCT = allogeneic hematopoietic stem cell transplantation; Cz Ph = Philadelphia Chromosome.

**Figure 2 medicina-59-00671-f002:**
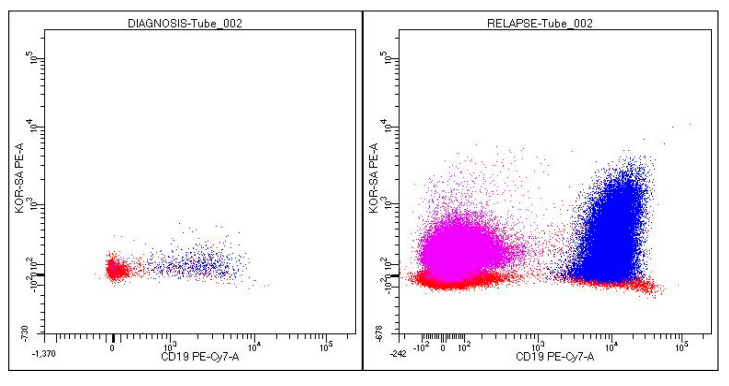
Immunophenotypic expression of KOR-SA at diagnosis versus relapse. In blue: CD19+ blasts (and CD34+). In violet: granulocytes as internal control (almost absent at diagnosis due to the hypocellular marrow sample and consequently few events).

## Data Availability

Not applicable.

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
