# Peer review of "The Philadelphia Chromosome, from Negative to Positive: A Case Report of Relapsed Acute Lymphoblastic Leukemia Following Allogeneic Stem Cell Transplantation"

_medicina, 2023, doi:10.3390/medicina59040671_

Round 1

Reviewer 1 Report

This case report of a Ph-positive acute lymphoblastic leukemia (ALL) relapsed after Ph negative ALL described by the Authors could be considered for publication after a thorough review

Ph-positive ALL that has relapsed after allogeneic stem cell transplantation for Ph-negative ALL is a rare, although possible, event. For this reason, it is important to describe the characteristics of the disease and the outcome. In this case, when possible, it could be useful to perform an extended molecular sequencing (e.g. NGS), to acquire all biological information. Moreover, when possible, it could be useful to re-analyze the cells of the diagnosis. 

The Authors should review the discussion: the hypothesis on the onset of Ph-positive after Ph-negative ALL are not reported in a proper form. Furthermore, they must support their hypothesis with more references, in particular when they describe "the dormant clone ... that escaped the immune control .."

The Authors did not describe the main characteristics of the transplant: conditioning regimen (chemo- or TBI-based), the GVHD prophylaxis, AB0 and CMV serology donor-recipient combination, stem cell source (marrow or peripheral blood), the infused cell dose, PMN and PLTs engraftment.

The detection method of chimerism (molecular, FISH), and source (marrow and/or peripheral blood) should be reported

Data on donor chimerism at time of relapse after transplantation are relevant for the interpretation of the biological evolution of the leukemia 

The future treatment plan  (second transplant or CAR-T cell treatment) should be reported in the "presentation" part, not in discussion 

CRD: is not a consistent or conventional abbreviation for complete remission 

Author Response

Dear Dr.,

First of all, I would like to apologize for not being able to reply to you sooner. I sent the article (case report) revised, I made additions to reach a number of more than 2500 words. I thank you for your understanding. I am waiting for your evaluation. I wish you all the best.

Reviewer 2 Report

The authors describe a rare entity of Ph-negative ALL relapsed with the emergence of Ph-positive clone. However, the report lacks critical information about the patient and the disease course as the details of the treatment protocol, the status of remission and MRD after induction, consolidation, etc, the level of disease at the time of transplant if any, details of the conditioning regimen, cell source, GVHD prophylaxis, chimerism at relapse.

The authors fail to discuss relevant literature or similar case reports in the introduction or discussion.

e.g. https://pubmed.ncbi.nlm.nih.gov/2752368/

The authors mentioned the complete eradication of Ph chromosome and even transcripts by PCR after only 14 days of TKI steroid therapy which is unusual, I would like the author to comment on this.

We would like to know if any consolidation therapy as DLI or second allo was employed and how long the patient is kept on TKI alone, the tolerability, any repeated bone marrow evaluation to confirm the patient is still in remission and BCR-ABL1 transcripts are still undetected.

The author mentions contamination from the donor as a possible reason for the presence of PH +ve chromosome at relapse in the discussion. Nonetheless, this is usually present at a very low level and not at this high level, specially when the bone marrow is full of blasts likely from donor origin. Another explanation would be missing the Ph +ve ALL clone at the diagnosis, and it may have contributed to persistent MRD disease before the transplant.

It would strengthen the manuscript to share figures from the chromosome analysis at diagnosis and relapse, also flow cytometry plots.

Author Response

(The authors gave the same response as above.)

Round 2

Reviewer 2 Report

I do not believe the authors addressed my previous comments.

Author Response

The authors describe a rare entity of Ph-negative ALL relapsed with the emergence of Ph-positive clone. However, the report lacks critical information about the patient and the disease course as the details of the treatment protocol, the status of remission and MRD after induction, consolidation, etc, the level of disease at the time of transplant if any, details of the conditioning regimen, cell source, GVHD prophylaxis, chimerism at relapse.

Answer:

  • The BFM 2000 protocol consisted of the following stages : induction (prednisone, vincristine, daunorubicin, PEG -asparaginase, intrathecal methotrexate), early intensification (cyclophosphamide, cytarabine, 6-mercaptopurine, intrathecal methotrexate) , consolidation (combination of dexamethasone, vincristine, vindensine HD-cytarabine, HD-methotrexate, cyclophosphamide, ifosfamide, PEG-asparaginase, etoposide, intrathecal therapy), and re-induction therapy (dexamethasone, vincristine, doxorubicin, PEG-asparaginase, cyclophosphamide, cytarabine, 6-thioguanine), followed by maintenance (6-mercaptopurine, methotrexate).
  • The patient achieved complete morphological marrow remission immediately after the induction phase. After the reinduction phase, i.e., 9 months after the start of treatment, the patient is still in complete morphologic remission, without CNS damage, but with the presence of minimal residual disease of less than 0.1% as determined by flow cytometry.
  • At this time, the patient underwent allo-HSCT from a matched related donor, knowing that MRD is the most important prognostic information and decision support for allocation to allogeneic hematopoietic stem cell transplantation
  • The procedure was successful, with no major complications and no graft-versus-host disease (GVHD). Chemotherapy-based myeloablative transplant conditioning regimens have been applied without total body irradiation (TBI) - for technical reasons. Six months after allo-HSCT, the patient was in complete remission, with negative MRD and 100% chimerism.

The authors fail to discuss relevant literature or similar case reports in the introduction or discussion.

e.g. https://pubmed.ncbi.nlm.nih.gov/2752368/

Answer:

  • First, I would like to state that this case report was the result of a personal attempt to find an answer to the question of why this young patient relapsed. I am well aware that relapse of acute lymphoblastic leukemia after bone marrow transplantation is not uncommon, but the relapse in this patient was rather atypical, with a cytogenetic change that was not present at diagnosis.
  • I tried to find out in the literature if such a relapse was reported, but I did not find it. At least for this category of patients, AYA, which has its own characteristics.
  • I was trying to understand at what stage of the development and treatment of the disease this chromosomal change occurred and whether there was anything I could have done as a doctor to prevent it?

The authors mentioned the complete eradication of Ph chromosome and even transcripts by PCR after only 14 days of TKI steroid therapy which is unusual, I would like the author to comment on this.

Answer:

  • Very quickly, after another week of treatment with Dasatinib, bone marrow evaluation already showed morphologic remission and Bcr-Abl transcript was no longer detected. It should be noted that the sample studied was from peripheral blood and the number of copies was reduced.

We would like to know if any consolidation therapy as DLI or second allo was employed and how long the patient is kept on TKI alone, the tolerability, any repeated bone marrow evaluation to confirm the patient is still in remission and BCR-ABL1 transcripts are still undetected.

Answer:

  • At this time, the patient is on treatment with Dasatinib 3 months after relapse after bone marrow transplantation, with maintenance of morphologic and molecular remission, and is awaiting CAR -T therapy, as a bridge to a second bone marrow transplantation.

The author mentions contamination from the donor as a possible reason for the presence of PH +ve chromosome at relapse in the discussion. Nonetheless, this is usually present at a very low level and not at this high level, specially when the bone marrow is full of blasts likely from donor origin. Another explanation would be missing the Ph +ve ALL clone at the diagnosis, and it may have contributed to persistent MRD disease before the transplant.

Answer:

  • Several hypotheses can be considered that underlie the relapse of the disease. One is that there was a dormant clone that could not be identified at diagnosis that escaped immune system control? Is this mutation acquired during the development of the disease or is it secondary to treatment? Could possible contamination by the donor also be considered? The fact that a myeloablative conditioning regimen was used without TBI may have favored relapse?
  • In view of all this, the question arises whether it would not be more effective to include TKIs in treatment in prophylactic doses and after certain clinical trials, starting from the diagnosis of ALL?

It would strengthen the manuscript to share figures from the chromosome analysis at diagnosis and relapse, also flow cytometry plots.

Answer:

        Fig. 1  Immunophenotypic expression of KOR-SA at diagnosis versus relapse

        In blue : CD19+ blasts (and CD34+). In violet: granulocytes as internal control

        (almost absent at diagnosis due to  the hypocellular marrow sample and consequently few events)

Round 3

Reviewer 2 Report

The authors made significant changes to the manuscript and It can be accepted in its current form.